# Protective Impact of Chitosan Film Loaded Oregano and Thyme Essential Oil on the Microbial Profile and Quality Attributes of Beef Meat

**DOI:** 10.3390/antibiotics11050583

**Published:** 2022-04-26

**Authors:** Abdul Basit M. Gaba, Mohamed A. Hassan, Ashraf A. Abd EL-Tawab, Mohamed A. Abdelmonem, Mohamed K. Morsy

**Affiliations:** 1Department of Food Hygiene and Control, Faculty of Veterinary Medicine, Benha University, Qaluobia 13736, Egypt; abdul.gaba@fvtm.bu.edu.eg (A.B.M.G.); mohamed.hassan@fvtm.bu.edu.eg (M.A.H.); 2Department of Quality Systems and Sustainability, Kalustyan Corporation, 855 Rahway Ave, Union, NJ 07083, USA; 3Department of Bacteriology, Immunology, and Mycology, Faculty of Veterinary Medicine, Benha University, Qaluobia 13736, Egypt; ashraf.awad@fvtm.bu.edu.eg; 4Agriculture Research Center, Central Lab of Residue Analysis of Pesticides and Heavy Metals on Food, Food Microbiology Unit, Cairo 12311, Egypt; doctormmm23@hotmail.com; 5Department of Food Technology, Faculty of Agriculture, Benha University, Qaluobia 13736, Egypt

**Keywords:** chitosan film, oregano oil, thyme oil, meat quality, food microbiology, antioxidants

## Abstract

Edible films and essential oil (EO) systems have the potency to enhance the microbial quality and shelf life of food. This investigation aimed to evaluate the efficacy of chitosan films including essential oils against spoilage bacteria and foodborne pathogens associated with meat. Antimicrobial activity (in vitro and in vivo) of chitosan films (CH) incorporated with oregano oil (OO) and thyme oil (TO) at 0.5 and 1% was done against spoilage bacteria and foodborne pathogens, compared to the control sample and CH alone. Preliminary experiments (in vitro) showed that the 1% OO and TO were more active against *Staphylococcus aureus* compared to *Escherichia coli* O157:H7 and *Salmonella* Typhimurium. In in vivo studies, CH containing OO and TO effectively inhibited the three foodborne pathogens and spoilage bacteria linked with packed beef meat which was kept at 4 °C/30 days compared to the control. The total phenolic content of the EOs was 201.52 mg GAE L^−1^ in thyme and 187.64 mg GAE L^−1^ in oregano. The antioxidant activity of thyme oil was higher than oregano oil. The results demonstrated that the shelf life of meat including CH with EOs was prolonged ~10 days compared to CH alone. Additionally, CH-OO and CH-TO have improved the sensory acceptability until 25 days, compared to the control. Results revealed that edible films made of chitosan and containing EOs improved the quality parameters and safety attributes of refrigerated or fresh meat.

## 1. Introduction

Food safety is a major global concern, with implications for both health and commerce [1]. Foodborne diseases are one of the global public health challenges that are affecting larger and larger segments of the population, particularly the elderly and people with immune system deficiencies [2]. Every year, approximately 179 million people become ill, 428,000 are hospitalized, and 6000 die in the USA as a result of foodborne pathogens, which cost the United States USD 15.6 billion [3].

Although beef meat has a specific composition and high nutritive value, the nonacid food category (pH > 4.5) encourages the two groups of microorganisms to grow such as spoilage bacteria including *Acinetobacter, Aeromonas, Br. thermosphacta, Enterobacteriaceae, Lactobacillus, Moraxella,* and *Pseudomonads* [4] as well as pathogens including *Escherichia coli* O157:H7, *Listeria monocytogenes, Staphylococcus aureus,* and *Salmonella* Typhimurium [5]. Moreover, it is susceptible to biochemical deterioration i.e., oxidation reactions [6], and changes in quality parameters i.e., discoloration, off-odors, off-flavors, and texture deterioration especially during storage [7].

Essential oils (EOs) have been considered an interesting alternative for food preservation [8]. They are confirmed as Generally Recognized As Safe (GRAS) as food additives [9]. Additionally, have several benefits such as antibacterial, antifungals, antioxidants, and anti-inflammatory [10]. Previous researchers found that EOs have antimicrobial activity against several microbial growths in different foods i.e., meats, meat products, fish, dairy products, vegetables, and fruits [11,12,13,14]. The antimicrobial capacity of EOs is due to several compounds such as *p*-cymene, carvacrol, thymol, and γ-terpinene [15].

Among of important EOs are oregano (*Origanum vulgare*) and thyme (*Thymus vulgaris*) oil which are used successfully in food processing and preservation [16,17] due to their antimicrobial and antioxidant activities. Both EOs are rich in phenolic i.e., carvacrol and thymol, while having lower contents of γ-terpinene and *p*-cymene [18]. The antimicrobial ability of phenolic is linked the to–OH group that causes leakage of cell membrane and damages bacteria [19,20]. EOs have been demonstrated to be active against both G^+^ and G^−^ microorganisms [21] and/or enhance the shelf life of foods [22]. Moreover, oregano and thyme oil have been utilized for flavoring meat, fish, and sauces [23,24].

Chitosan (CH) is a derived polysaccharide from chitin found in shrimp and/or crustacean shells [25]. CH-based films have a variety of physical-functional attributes that makes them technically appropriate to apply in food contact materials [26]. Previous research has demonstrated that CH is unique in meat or meat products because of the good barrier to oxygen and moisture as well as being non-toxic, biodegradable, and cheap [27]. CH-based films have also enhanced the microbial quality of chilled meat [28], pork patties [29], turkey meat [30], and chicken meat [31]. The aim of the present study was to (i) evaluate the antimicrobial activity of chitosan-based film incorporating EOs (oregano and thyme) on foodborne pathogens and background bacteria, and (ii) evaluate the impact of chitosan films with EOs on quality parameters of beef meat.

## 2. Results and Discussion

### 2.1. GC/MS-MS Fingerprint and Characterization of Essential Oil

The thyme and oregano were planted and extracted on a commercial scale, the oil percentage was found at 2.2% yield for oregano and 1.05% for thyme. As shown in Figure 1 and Figure 2, the GC/MS composition fingerprint of oregano oil (OO) and thyme oil (TO) were evaluated. A total of 42 and 34 compounds were identified (act ~95–99% of the total components) in OO and TO, respectively. Major compounds in OO were carvacrol (58.30%), linalool (9.09%), γ-terpinene (6.01%), p-cymene (4.31%), β-bisabolene (3.74%), thymol (3.49%), caryophyllene (1.67%), myrcene (1.63%), and thujene (1.27%). While in TO the main compounds were thymol (56.16%), *p*-cymene (11.67%), *γ*-terpinene (8.11%), and carvacrol (5.47%), caryophyllene (5.16%), *γ*-terpinene (1.59%), and borneol (1.48%). The study conducted by Govaris et al. [32] found that the major compounds in OO, including carvacrol, *p*-cymene, thymol, and γ-terpinene were 80.15, 5.18, 4.82, and 0.77%, respectively. Other research by Nikolić, Glamočlija, Ferreira, Calhelha, Fernandes, Marković, Marković, Giweliand, and Soković [10] showed that TO have thymol and *p*-cymene 49.10 and 20.01%, respectively. The variation in EOs compounds depends on some factors including varieties, harvesting seasons, and geographical sources. These findings are consistent with those previously reported in [33,34].

### 2.2. Phenolics and Antioxidants Activity

As shown in Table 1, the phytochemical (total phenolic: TP) in essential oil was evaluated. The results revealed that TP content in thyme and oregano oil was 201.52 and 187.64 mg GAE L^−1^, respectively. Mutlu-Ingok et al. [35] found that the phenolic content of thyme and oregano oil was 193 and 163 mg GAE L^−1^, respectively. The variation in TP content may be due to the variety or extraction method. Additionally, Table 1 shows the antioxidant activity of the essential oils (DPPH). The radical scavenging capacity of thyme oil is more than oregano oil. A study by Mutlu-Ingok, Catalkaya, Capanogluand, and Karbancioglu-Guler [35] found that thyme and oregano oil have a 56.2 and 53.4%, respectively, scavenging effect on DPPH and marked reducing power.

### 2.3. Antibacterial Activity of CH Films Containing OO and TO Using Plate Overlay Assays (In Vitro)

The antibacterial influence of chitosan films incorporating EOs such as oregano (OO) and thyme (TO) was evaluated against various pathogenic microorganisms such as *E. coli* O157:H7, *S. aureus*, and *S.* Typhimurium (Table 2) in plate overly assays. Results showed that the oregano oil (OO) at a concentration of 1% (*w*/*v*) inhibited the growth of pathogens bacteria such as *S. aureus, S.* Typhimurium, and *E. coli* O157:H7 more than the concentration of 0.5% OO. While the chitosan films that incorporated oregano oil (CH-OO 1%) were more active against selected foodborne pathogens. Additionally, it was found that thyme oil (TO; 1% *w*/*v*) is active against *E. coli* O157:H7 (20 mm), *S. aureus* (50 mm), and *S.* Typhimurium (21 mm), whereas the chitosan films including TO (CH-TO 1%) were highly inhibited the foodborne pathogens. Generally, TO and/or CH-TO have higher activity against foodborne pathogens than OO and/or CH-OO. These findings are generally consistent with those reported in other studies [17,36,37].

The antimicrobial impact of EOs is due to carvacrol and thymol compounds that are in OO and TO, respectively. The EOs’ mode of action against foodborne bacteria is thought to be a result of the disruption of the cell membrane [38]. Because carvacrol and thymol are generally hydrophobic, it targets the bacterial cell membrane, causing leakage of cytoplasmic constituents, and making them more permeable [2], as well as causing leakage of phosphate ion in *S. aureus* and *P. aeruginosa* [39]. One study by Simirgiotis et al. [40] found that OO has affected the bacterial cell wall and caused leakage of intracellular contents. Other studies demonstrated that the carvacrol increases the permeability of bacterial strains’ cytoplasmic membrane to potassium ions and proteins, that is led to bacterial death [41]. Moreover, these EOs may penetrate of the cell membrane, disrupting, and interacting between fatty acids, phospholipids, and proteins, as well as preventing ATP production, interacting with other organelles, and causing cell death [42]. Therefore, the use of EOs to prevent the growth of pathogenic microorganisms may represent a viable alternative to the use of chemical preservatives and the preparation of food free of synthetic additives [43].

### 2.4. Impact of CH Films Containing OO and TO on Background Microflora of Beef Meat (In Vivo)

#### 2.4.1. Psychrophilic Bacteria

As shown in Figure 3a, the impact of CH films including OO and TO on psychrophilic bacteria growth up to 30 days at 4 ± 1 °C in beef samples was done. Results illustrated that CH-OO (0.5 and 1%; *w*/*v*) reduced the growth of psychrophilic bacteria from 4.4 log CFU g^−1^ to 2.3 and 1.6 log CFU g^−1^, respectively, compared to the control and chitosan alone (7.8 and 7 log CFU g^−1^) during storage up to 30 days. That is due to the carvacrol compound which is able to damage the bacterial membrane, especially G^+^ bacteria [44]. Moreover, carvacrol makes changes in fatty acids, decreasing the proton outflow and running out of the cytoplasm of the bacterial cell [45]. However, the CH films incorporating TO controlled the psychrophilic bacteria population to 2 and 1.2 log CFU g^−1^ (0.5% and 1%; *w*/*v*), respectively, for up to 30 days of storage. The previous data revert to the capability of thymol to lose the cell wall role, increasing the permeability and depolarization [46]. Additionally, it prevents ATP production, coagulating the protein, and losing a vital role [47].

#### 2.4.2. Pseudomonas Bacteria

Figure 3b shows that CH-OO decreased the *pseudomonas* growth on beef to 1 log CFU g^−1^ (0.5%; *w*/*v*) and 0.6 log CFU g^−1^ (1%; *w*/*v*) compared to the control sample (6 log CFU g^−1^). This reduction is due to the carvacrol compound phenol monoterpene, which can interact and penetrate the outer cell wall of bacteria, causing leakage of intracellular proteins, potassium ions, and decreased ATP production [48]. One study by Maggini et al. [49] showed carvacrol has low effects on the membrane and mitochondria of bacteria, as well as losing a vital roles’ membrane. Additionally, CH-TO has the same inactivation impact on *pseudomonas* population up to 1 log CFU g^−1^ (0.5%; *w**/v*) and 0.3 log CFU g^−1^ (1%; *w**/v)* when compared to the control. The inactivation of bacterial growth reverts to thymol, which penetrates the cytoplasm, enzyme denaturation, and protein coagulation associated with the prevention of cellular functions [50].

#### 2.4.3. Lactic Acid Bacteria (LAB)

Figure 3c shows the LAB in meat treated with oregano and thyme oils. The obtained results indicated CH-OO has the ability to inactivate the LAB load till 1.7 log CFU g^−1^ (0.5%; *w**/v*) and 1.5 log CFU g^−1^ (1%; *w**/v*) compared to control (6.9 log CFU g^−1^). On the other hand, CH-TO inhibits the LAB population to 1.4 log CFU g^−1^ (0.5%; *w**/v*) and 1 log CFU g^−1^ (1%; *w**/v*) during storage for up to 30 days. The reduction of LAB may be due to some reasons, i.e., hydroxyl groups of carvacrol can destroy the cell wall and reduce the membrane roles [51], running out of ions, ATP, and nutrients through the membrane [52], or increasing the fluidity of bacterial cell membrane and this increase in the leakage leads to cell death [53].

### 2.5. Impact of CH Films Incorporating OO and TO on Foodborne Pathogens of Beef Meat (In Vivo)

According to the findings of the plate overlay assays, chitosan films containing 0.5% OO, 1% OO, 0.5% TO, and 1% TO were assessed for their long-term efficacy as an antimicrobial against *E. coli* O157:H7, *S. aureus,* and *S.* Typhimurium associated with raw vacuum-packaged beef under at 4 °C for up to 30 days. Figure 4 reveals the antimicrobial activity of chitosan films with and/or without the addition of essentials oils against *E. coli* O157:H7 on raw beef. It was discovered that the bacterial populations grew indefinitely continuously during the 30 days of storage when treated with the CH films (without EO). On the contrary, a chitosan-based film incorporating OO and TO (1%; *w**/v*) showed a decrease of ~2.3 and 1 log unit in populations of *E. coli* O157:H7 until the end of the challenge study (Figure 4a). *S. aureus* was sensitive to the EOs antimicrobial films, as indicated by the rate of reductions population which ranged from 3 to 4 log_10_ CFU/g on raw beef, compared to the control, and remained constant until the end of the challenge study (Figure 4b). A similar trend of reductions of *S.* Typhimurium in raw beef that was refrigerated and vacuum packaged was noted for the duration of the experiment (Figure 4c). The inactivation of pathogens is due to carvacrol and thymol, as well as phenolic compounds which prevent cellular functions, penetrate the cytoplasm, prevent ATP production, and lose vital roles [50,51]. These findings clearly reveal that essential oils included in a chitosan biopolymer can be inactivated foodborne pathogens in fresh meat for up to 30 days.

### 2.6. Impact of CH Films Incorporating OO and TO on Meat Color during Storage Time

The color characteristics i.e., L* value (lightness), a* value (redness), and b* value (yellowness) of control samples and treated samples with CH-OO and CH-TO at (0.5% and 1% *w*/*v*) were determined during the storage period. As shown in Figure 5, the L* value was decreased gradually during storage time for all samples as follows: control samples from 36.1 to 33.5, samples with CH-OO (0.5%; *w*/*v*) were 44.1 to 35.5, and samples with CH-OO (1%; *w*/*v*) 45.3 to 42.8. While the a* value was increased gradually during the storage period with meat samples as follows: control samples from 12.1 to 14.1, samples with CH-OO (0.5% *w*/*v*) 10.3 to 13, and samples with CH-OO (1%; *w*/*v*) 10 to 12.7. The b* value of untreated meat was slightly decreased during the storage period from 11.8 to 10.7, however, the treated samples with CH-OO (0.5% and 1%; *w*/*v*) were increased. A similar direction of color change with CH-TO at 0.5% and 1% (*w/v*). Results demonstrated that CH-OO and CH-OT have a significant impact on meat color parameters and enhance their acceptability. The results are consistent with those previously reported by Yemiş and Candoğan [54].

### 2.7. Sensory Evaluation

The sensory response of impacts of coating treatments on meat samples’ color, odor, and overall acceptability was evaluated and is displayed in Figure 6. The data demonstrated no significant difference in sensory parameters between the meat treatments at time zero of storage (*p* ≥ 0.05). The organoleptic assessment was performed for 10 days for the control and chitosan, 20 days for CH-OO and CH-TO at 0.5% (*w*/*v*), and 25 days for CH-OO and CH-TO at 1% (*w*/*v*) because of the unpleasant odor, which most likely indicates spoilage, which would lead to consumer rejection. Generally, the color scores for all meat treatments decreased as storage time progressed. The control sample showed the lowest score and was rejected in the early stage of storage. While the addition of chitosan-based films containing OO and TO at both levels (0.5% and 1%; *w/v*) had no negative impact on the color of the meat. The changes in odor were also increased in all treatments with increased storage time, due to lipid oxidation of meat and protein deterioration. The meat samples treated with essential oils were quite stable for 25 days, compared to the control sample for 10 days. In general, the overall acceptability of the control samples decreased due to the dark red color and off odor, but the meat coated with active packaging was lightly dark red or cherry red and had a pleasant odor. These results confirmed that chitosan films and essential oil improved the sensory acceptability of meat, which may be due to EOs’ antibacterial and antioxidant properties [55].

## 3. Materials and Methods

### 3.1. Raw Materials

Two aromatic plants i.e., oregano (*Origanum vulgare*) and thyme (*Thymus vulgaris*) were planted and collected from Kalustyan farms in Turkey and Egypt in 2018. Raw beef was bought from a native market in New York, USA. The meat was delivered within 10 min to the laboratory in an icebox. Chitosan (75% deacetylation) was obtained from Sinopsin group chemical co., Beijing, China. Magnesium sulfate anhydrous, sodium carbonate, acetic acid, Gallic acid xanthan gum, glycerol, tween, DPPH, Folin–Ciocalteu, and ethanol 95% were supplied from Sigma-Aldrich (Louis, MO, USA).

### 3.2. Microorganisms

Three pathogen strains i.e., *E. coli* O157:H7 (ATCC 25922), *S. aureus* (ATCC 6538), and *S.* Typhimurium (ATCC 14028) were housed in Food Microbiology Lab in Kalustyan Company. All bacteria were kept and cultivated on Tryptic Soy Agar (TSB, Difco Laboratories) at 37 ± 1 °C for 24 h. Then, the bacterial cultures were propagated in Tryptic Soy Broth (TSB, Difco) and incubated at 37 ± 1 °C/16 h before experiments.

### 3.3. Essential Oil Extraction

Fresh leaves of oregano (*Origanum vulgare*) were collected from the Denizli region in Turkey and thyme leaves (*Thymus vulgaris*) from the Beni Suef region in Egypt. The leaves were dehydrated by sun-drying and cracked using a knife mill (Model # D90, Cimbria, Denmark). The dried leaves were placed in 4 mm polyliner food-grade bags and transported to an essential oil distillation facility (Albkalusyan, Albania). The essential oils were extracted from leaves by distillation method as Stratakosand Koidis [56], with minor changes. Briefly, the dried leaves of oregano and thyme were placed into a distillation tank. The steam was generated and injected (Boiler, Babcock, and Wilcox) for 5–6 h. The essential oils were condensed and received in the aluminum flask, then filtrated using magnesium sulfate anhydrous and stored under cooling conditions. The yield of oil was calculated from Equation (1):(1)$$Oil\;Yield=\frac{Oil\;\left(Kg\right)}{Leaves\;in\;\left(KG\right)}*100$$

### 3.4. GC/MS-MS Fingerprint

The composition of oregano and thyme oils was done by GC/MS according to Morsy, Khalaf, Sharoba, El-Tanahi, and Cutter [1]. The GC/MS equipment was an Agilent 5977A mass elective detector, and a quadrupole Electron Ionization mass analyzer. It had an HP–5MS 5% column with 30 m × 0.25 mm, 0.25 μm thicknesses as the stationary phase. Helium (carrier gas) at 0.95 mL min^−1^ flow rate. The initial temperature was 60 ± 1 °C/2 min, increased to 250 ± 1 °C at 4 °C min^−1^. The temperatures of the injector and the interface were adjusted at 250 °C and 280 °C, respectively. The temperatures of the ion source and detector were 230 and 150 °C, respectively. A sample was injected with a split ratio of 1:100. The compound description was made from the mass spectra of compounds using the Agilent library as well as the Wiely and the 9 nist11 libraries [57].

### 3.5. Chitosan Film Preparation

The chitosan (CH) aqueous was appointed by dissolving 15 g CH (75% deacetylation; Sinopsin group chemical co., China) in 1000 mL acetic acid (0.5%; *v/v*). Glycerol (0.2%; *w/v*) as plasticizer, xanthan gum (0.1%; *w/v*) as stabilizer, and tween 80 (0.5%; *w/v*) as a surfactant were added. When all the ingredients were fully dissolved (4 h), the CH aqueous was sterilized at 121 ± 1 °C/15 min and left to cool down to 25 ± 1 °C. The OO and TO were added in concentrations 0.5% and 1% (*w/v*) and gently mixed. After that, the films were poured into sterile Petri plates (VWR) and left to dry ~22 h in a laminar airflow cabinet at 25 ± 1 °C. The chitosan films were kept in the refrigerator until use [55].

### 3.6. Plate Overlay Assays with Essential Oils

The antimicrobial ability of EOs was done by a plate overlay assay method [58]. TSA plates were layered with ten microliters of soft agar seeded with 100 μL of broth culture of *E. coli* O157:H7, *L. monocytogenes, S. aureus*, or *S.* Typhimurium. The density was ~6 log_10_ CFU mL^−1^ in the overlay. A total of 20 μL of oregano and thyme essential oils were put directly onto seeded lawns and left 5 min under the laminar air hood. Plates were incubated for 24 h at 37 ± 1 °C before being scored for inhibition zones. To evaluate the ability of the chitosan (CH) films including essential oils, antimicrobial CH films were cut aseptically into 1 cm × 1 cm and tested against target bacteria using the plate overlay method described previously [1].

### 3.7. Challenge Studies

#### 3.7.1. Trial I (Natural Microbiota)

In this trial, the antimicrobial ability of CH films incorporated EOs was performed on meat contaminating natural microbiota. Using a deli slicer machine, the raw meat was cut into 1 mm slices (Globe, Ohio, USA), then slices were cut into 5 cm × 5 cm sections. The meat slice was coated with CH films (5 cm × 5 cm) containing essential oils. All samples were transferred into a standard vacuum-packaging machine (Ultravac 250 machine, Louis, MO, USA). The control sample (uncoated) and chitosan film without antimicrobial properties were performed. Vacuum packaged treatments were stored for 30 days at 4 ± 1 °C. The samples were checked at intervals of 0, 5, 10, 15, 20, 25, and 30 days.

#### 3.7.2. Trial II (Pathogens Bacteria)

In the second trial, the raw meat was cut into pieces (5 × 5 cm) under aseptic conditions at a laminar airflow cabinet. The meat samples were exposed to ultraviolet light on the surface (UV; 250 nm) for 15 min to reduce natural microflora. The samples were aseptically inoculated by diluted cultures of *E. coli* O157:H7, *S. aureus*, and *S.* Typhimurium to obtain ~6 log_10_ CFU/cm^2^ on the surface and left for 15 min to make cell attachment possible. The meat slice was coated with CH films 5 cm × 5 cm incorporating essential oils. The samples are put into a standard vacuum-packaging machine. The control sample (uncoated) and chitosan film without antimicrobial properties were analyzed. Vacuum-packaged treatments were stored for 30 days at 4 ± 1 °C. The samples were examined for the presence of any remaining microbial populations on days 0, 5, 10, 15, 20, 25, and 30.

### 3.8. Microbiological Analyses

The vacuum-packed samples were opened and aseptically put into a filtered stomacher bag (Rockland, MA, USA) containing 25 mL of 0.1% buffered peptone water (BPW, Difco), homogenized (Seward 400 Stomacher, West Sussex, England), and the filtrate was collected. Serial dilutions were done and 100 μL spread plated (duplicate) onto Sorbitol MacConkey Agar (SMAC, Difco) for *E. coli* O157:H7, Baird–Parker agar (BP, Difco) for *S. aureus*, Xylose Lysine Deoxycholate Agar (XLD, Difco) for *S.* Typhimurium, Plate Count Agar (TBC, Difco) for psychrophilic bacteria, *Pseudomonas* agar (PA, Difco) for *pseudomonas*, MRS agar, and Oxoid for Lactic acid bacteria to determine the number of presence any remaining cells. After 24 to 48 h of incubation at 37 ± 1 °C, colonies were counted, while psychrophilic bacteria were counted after 5 d at 7 °C, populations are expressed as log_10_ CFU/cm^2^ [1].

### 3.9. Phenolics and Antioxidants Activity

The total phenolics (TP) of the essential oils (EOs) were investigated using the Folin–Ciocalteu method according to Viuda-Martos et al. [59]. Briefly, 200 µL of ethanolic solution of essential oil, mixed gently with Folin–Ciocalteu reagent (0.5 mL for 3 min) and 2 mL sodium carbonate (20%; *w*/*v*) was added. The mixture was left in the dark for 5 min at 50 ± 1 °C, and absorbance was done at 750 nm. TP was exposed as mg of gallic acid equivalent (GAE) per liter of essential oil. The DPPH radical scavenging capacity of EO was performed by the method reported by Liu et al. [60]. Briefly, 200 µL of EO was mixed with a 3.8 mL DPPH solution and incubated at 25 ± 1 °C in the dark condition for 1 h, and measured at 517 nm. Data were exposed as IC_50_ (inhibition, %) using Equation (2):% inhibition = (*A*_C (0)_ − *A*_EO (t)_/*A*_C (0)_) × 100(2)
where *A*_C (0)_ is the absorbance of the control at *t* = 0 min and *A*_EO (t)_ is the absorbance of the essential oil at the end of the incubation time.

### 3.10. Color Measurement

The color value of untreated and treated beef with essential oils was measured using a spectrophotometer CM-508d (Minolta Corp., Ramsey, Columbia, MI, USA). The values of *L*^∗^, *a*^∗^, and *b*^∗^ were measured. A standard white tile was used to calibrate the instrument.

### 3.11. Sensory Evaluation

A twelve-trained person from the Kalustyan Company assessed the beef meat treatments during the chilled storage. Samples were put into covered plates with coded-3-digit numbers. The panelists were asked to assess the treatments using the hedonic scale method (five-point) for color, odor, and overall acceptability with scores starting at 5 for excellent and 1 for unacceptable [61].

### 3.12. Data Analysis

This study’s experiments were carried out in triplicate using three samples in treatment, while the sensory evaluation was done with 12 panelists. Factorial design ANOVA with two factors followed six treatments (control, CH, CH-OO 0.5%, CH-OO 1%, CH-TO 0.5%, and CH-TO 1%), and storage time with seven points at 0, 5, 10, 15, 20, 25, and 30 days were applied. For the challenge study, the remaining bacteria were statistically analyzed using ANOVA with a significance level of *p* < 0.05 using SPSS, version 19 (IBM; Armonk, NY, USA) according to Steel et al. [62]. Tukey’s multiple comparison tests at *p* < 0.05 means were done.

## 4. Conclusions

In conclusion, the findings of this study indicate that oregano oil (OO) and thyme oil (TO) have highly bioactive compounds. The OO and TO incorporate into chitosan films were effective against pathogenic bacteria like *E. coli* O157:H7, *S. aureus*, and S. Typhimurium in plate overlay assays. In in vivo studies, CH containing OO and TO effectively inhibit the three pathogens and spoilage bacteria linked with beef meat kept at 4 °C/30 days, as compared to the control. The shelf life of CH-OO and CH-TO-treated meat was extended at least 10 days compared to that of CH alone and the control. Additionally, the CH-OO and CH-TO have improved the sensory acceptability until 25 days compared to the control. The findings revealed that CH-based films and contained EOs may enhance the quality and safety properties of fresh or chilled meat.

## Figures and Tables

**Figure 1 antibiotics-11-00583-f001:**
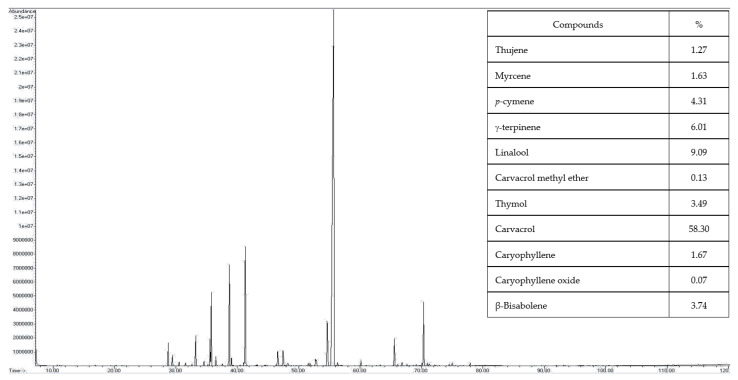
Oregano essential oil chromatogram by GS/MS/MS.

**Figure 2 antibiotics-11-00583-f002:**
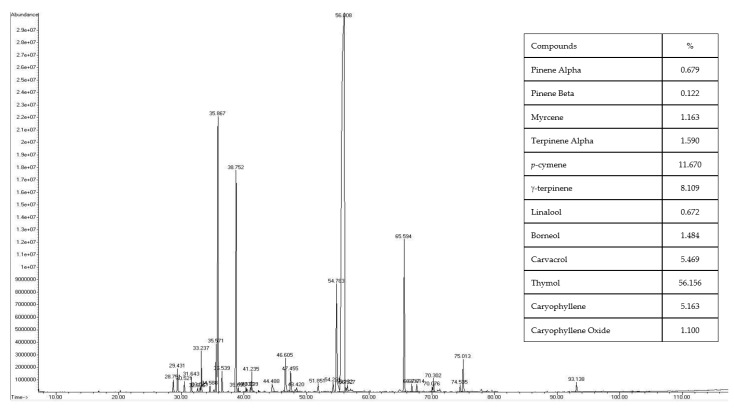
Thyme essential oil chromatogram by GS/MS/MS.

**Figure 3 antibiotics-11-00583-f003:**
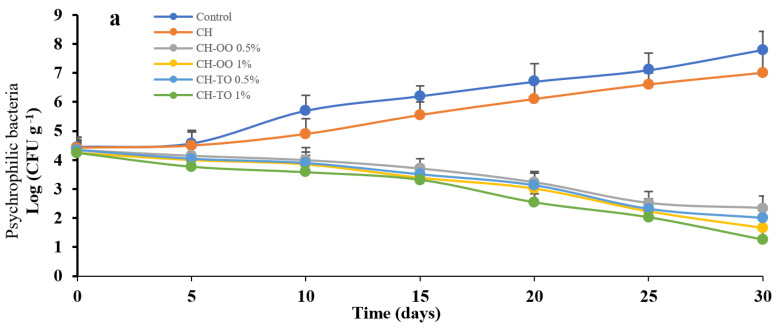
Antimicrobial activity of chitosan (CH) film incorporated oregano (OO) and thyme (TO) essential oils on Psychrophilic bacteria (**a**) *Pseudomonas* (**b**), and lactic acid bacteria (**c**) in the top round of beef during storage time. Error bars indicate standard deviation, (n = 3). Control: without film and oil; CH: chitosan; CH−OO 0.5%: chitosan+ oregano oil 0.5%; CH−OO 1%: chitosan+ oregano oil 1%; CH−TO 0.5%: chitosan+ thyme oil 0.5%; CH−TO 1%: chitosan+ thyme oil 1%.

**Figure 4 antibiotics-11-00583-f004:**
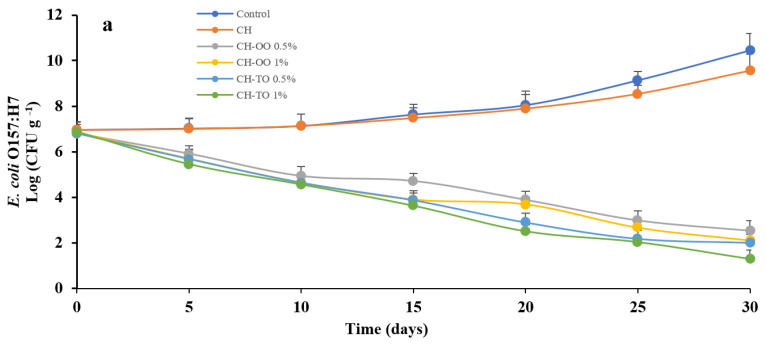
Antimicrobial activity of chitosan (CH) film incorporated oregano (OO) and thyme (TO) essential oils on *E. coli* O157:H7 (**a**) *S. aureus* (**b**), and *S.* Typhimurium (**c**) in the top round of beef during storage time. Error bars indicate standard deviation, (n = 3). Control: without film and oil; CH: chitosan; CH−OO 0.5%: chitosan+ oregano oil 0.5%; CH−OO 1%: chitosan+ oregano oil 1%; CH-TO 0.5%: chitosan+ thyme oil 0.5%; CH−TO 1%: chitosan+ thyme oil 1%.

**Figure 5 antibiotics-11-00583-f005:**
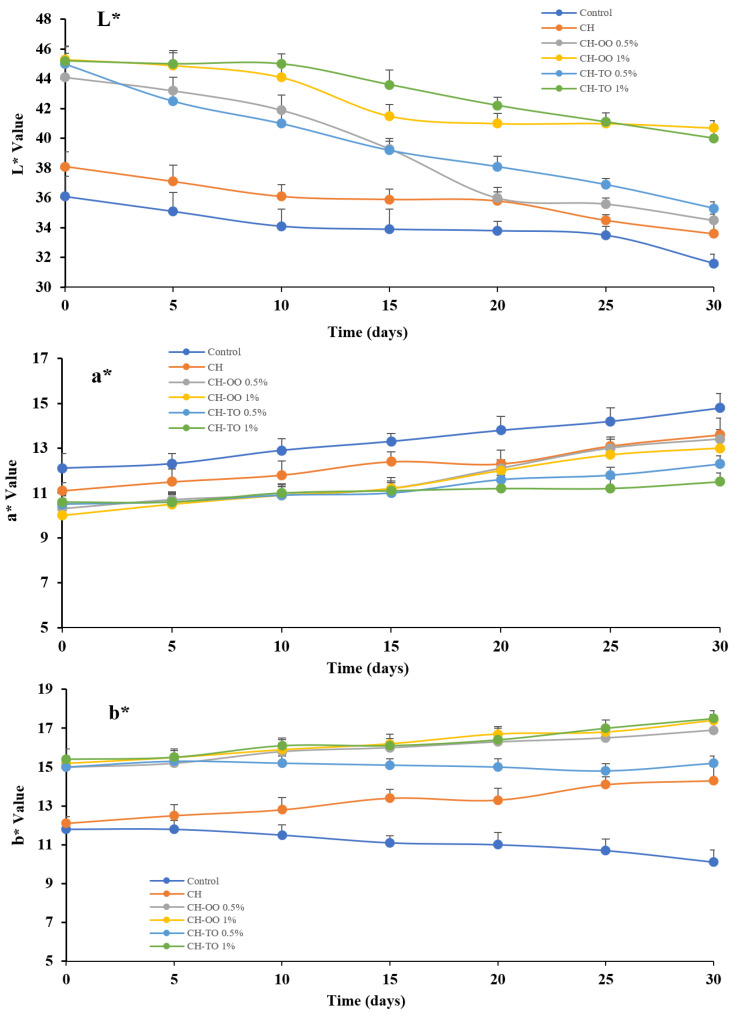
Impact of chitosan (CH) film incorporated oregano (OO) and thyme (TO) essential oils on beef color parameters i.e., L* value, a* value, and b* value during storage time. Error bars indicate standard deviation, (n = 3). Control: without film and oil; CH: chitosan; CH−OO 0.5%: chitosan+ oregano oil 0.5%; CH−OO 1%: chitosan+ oregano oil 1%; CH−TO 0.5%: chitosan+ thyme oil 0.5%; CH−TO 1%: chitosan+ thyme oil 1%.

**Figure 6 antibiotics-11-00583-f006:**
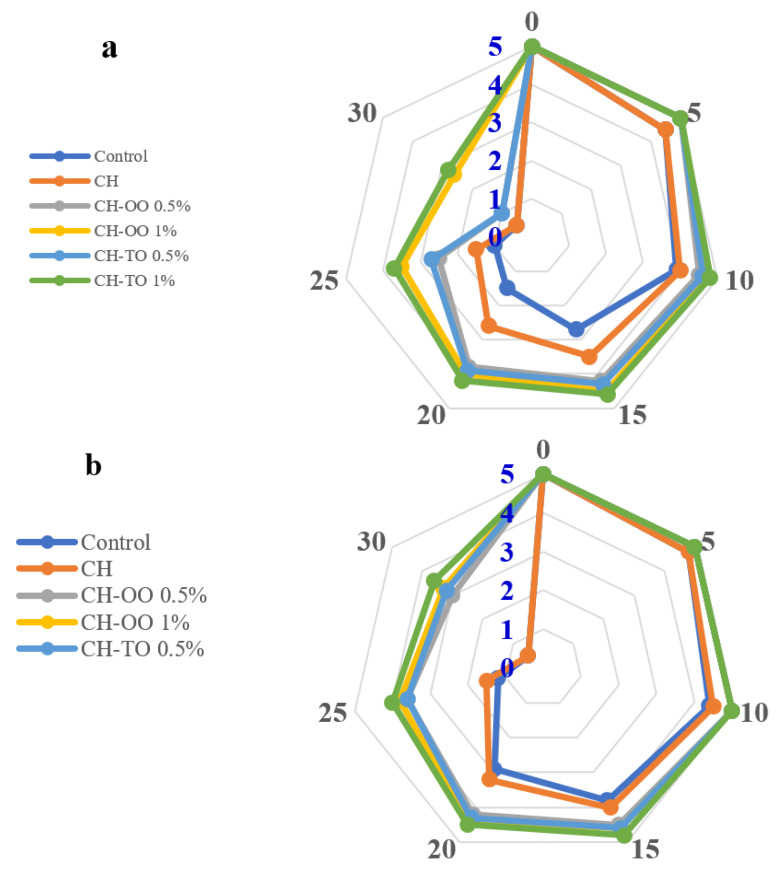
Sensory evaluation parameters i.e., color (**a**), odor (**b**), and overall acceptability (**c**) of beef meat with chitosan-based film incorporated oregano (OO) and thyme (TO) essential oils during storage at 4 °C (n = 12). Control: without film and oil; CH: chitosan; CH−OO 0.5%: chitosan+ oregano oil 0.5% CH−OO 1%: chitosan+ oregano oil 1%; CH−TO 0.5%: chitosan+ thyme oil 0.5%; and CH-TO 1%: chitosan+ thyme oil 1%.

**Table 1 antibiotics-11-00583-t001:** Total phenolic (TP) and radical scavenging activity of thyme and oregano essential oil (mean ± SD, n = 3).

Sample	TP (mg GAE * L^−1^ Sample)	IC_50_ Inhibition (%)
Thyme essential oil	201.52 ± 1.67 ^a^	58.44 ± 0.83 ^a^
Oregano essential oil	187.64 ± 1.65 ^b^	54.58 ± 0.62 ^b^

^a,b^ no significant difference between any two means “in the same column” with the same superscript letter (*p* ≥ 0.05). * Phenolics as gallic acid equivalents.

**Table 2 antibiotics-11-00583-t002:** Antibacterial activity of chitosan-based films containing oregano and thyme essential oils against foodborne pathogens (mm) (mean ± SD).

Bacterial Strains	Antimicrobial Activity (mm)							
	Oregano Oil				Thyme Oil			
	OO 0.5%	OO 1%	CH-OO 0.5%	CH-OO 1%	TO 0.5%	TO 1%	CH-TO 0.5%	CH-TO 1%
*E. coli* **O157:H7**	15 ± 1.17 ^cE^	21 ± 1.26 ^cB^	18 ± 1.11 ^bC^	24 ± 1.03 ^bA^	16 ± 1.15 ^bD^	20 ± 1.35 ^bB^	18 ± 1.03 ^bC^	23 ± 1.15 ^bA^
*S. aureus*	30 ± 1.45 ^aG^	45 ± 1.33 ^aD^	32 ± 1.32 ^aF^	47 ± 1.5 ^aC^	41 ± 1.6 ^aE^	50 ± 3.65 ^aB^	44 ± 1.43 ^aD^	52 ± 1.85 ^aA^
*S.* **Typhimurium**	17 ± 1.11 ^bE^	23 ± 1.33 ^bB^	19 ± 1 ^bD^	25 ± 1.16 ^bA^	16 ± 1.24 ^bE^	21 ± 1.25 ^bC^	19 ± 1.37 ^bD^	24 ± 1.22 ^bA^

OO: oregano oil; TO: thyme oil; CH-OO 0.5%: chitosan+ oregano oil 0.5%; CH-OO 1%: chitosan+ oregano oil 1%; CH-TO 0.5%: chitosan+ thyme oil 0.5%; CH-TO 1%: chitosan+ thyme oil 1%. ^a–c^ There are no significant differences between any two means “in the same column” with the same superscript small letter (*p* > 0.05). ^A–G^ There are no significant differences between any two means “in the same row” with the same superscript capital letter (*p* > 0.05).

## Data Availability

Data are contained within the article.

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
