# Peer review of "Protective Impact of Chitosan Film Loaded Oregano and Thyme Essential Oil on the Microbial Profile and Quality Attributes of Beef Meat"

_antibiotics, 2022, doi:10.3390/antibiotics11050583_

Round 1

Reviewer 1 Report

  1. Page 3 and Page 4, “Figure 1: ” and “Figure 2: ”.
  • The colon should be changed to a period.

  1. Page 5, Table 2.
  • Should "CH-OO 0.5%andCH-OO 1%" in the last two columns be "CH-TO 0.5% and CH-TO 1% "?

  1. Page 5, “Also, it was found that thyme oil (TO; 1% w/v) is active against E. coli O157:H7 (20 mm), S. aureus (50 mm), and S. Typhimurium (21 mm), whereas the chitosan films including OO (CH-OO 1%) were highly inhibited the foodborne pathogens. ”.
  • This sentence does not agree with the Table2.

  1. 4. Page 9,6,“Results demonstrated that e CH-OO and CH-OT have a significant impact on meat color parameters and enhancing their acceptability. The results are consistent with those previously reported by YemiÅŸ and CandoÄŸan [62]. ”.
  • Please check this sentence for tense and extra letters.

  1. 5. The uppercase and lowercase letters of bacteria names should be consistent in text,table, and figure.

Author Response

Reviewer # 1

  • Page 3 and Page 4, “Figure 1: ” and “Figure 2: ”. The colon should be changed to a period.

We appreciate the time spent on the review of our manuscript and for your criticisms and critiques of the reviewed manuscript. Ok, done.

  • Page 5, Table 2. Should "CH-OO 0.5%andCH-OO 1%" in the last two columns be "CH-TO 0.5% and CH-TO 1% "?

We agree. Ok, done.

  • Page 5, “Also, it was found that thyme oil (TO; 1% w/v) is active against E. coli O157:H7 (20 mm), S. aureus (50 mm), and S. Typhimurium (21 mm), whereas the chitosan films including OO (CH-OO 1%) were highly inhibited the foodborne pathogens. ”.This sentence does not agree with the Table2.

Ok, reviewed.

  • Page 9“Results demonstrated that e CH-OO and CH-OT have a significant impact on meat color parameters and enhancing their acceptability. The results are consistent with those previously reported by YemiÅŸ and CandoÄŸan [62]. ”.

Please check this sentence for tense and extra letters.

We agree. Ok, done.

The uppercase and lowercase letters of bacteria names should be consistent in text,table, and figure.

Ok, done.

Reviewer 2 Report

The paper “Protective Impact of Chitosan Film Loaded Oregano and Thyme Essential Oil on the Bacterial Diversity and Quality At-tributes of Beef Meat” analyse the efficacy, both in vitro and in vivo, of two essential oils /Tyme and Oregano) on the growth of three foodborne pathogens and other microbial groups, mainly spoilage microorganisms, such as lactic acid bacteria in beef meat. Preliminarily the essential oils were characterized. Moreover, the effect of the essential oils on the colour and the sensory characteristics of beef meat was evaluated.

The paper is interesting; however, some changes are needed.

No line numbers are present, thus making difficult to precisely address the comments.

In the title, I would replace “bacterial diversity” with “microbial profile”.

English must be deeply improved as many mistakes (both grammatical and syntactic) are present. Just as an example, in many sentences verbs are missing: “The antimicrobial capacity of the EOs IS due to several compounds like p-cymene, carvacrol, thymol, and γ-terpinene”. I would suggest a revision by a native speaker.

The abstract is very confusing and not self-supporting. Both the aims of the study and the experimental plan must be clarified.

In the in vitro tests, how was the cut-off for the inhibition zones diameter determined?

I can’t find the description of the methods for the determination of phenolics and antioxidant activity (chapter 2.2 in the results section).

Author Response

Reviewer # 2

  • No line numbers are present, thus making difficult to precisely address the comments.

We appreciate the time spent on the review of our manuscript and for your criticisms and critiques of the reviewed manuscript. Excuse me, now included.

  • In the title, I would replace “bacterial diversity” with “microbial profile”.

We agree. Ok, done.

  • English must be deeply improved, as many mistakes (both grammatical and syntactic) are present. Just as an example, in many sentences verbs are missing: “The antimicrobial capacity of the EOs IS due to several compounds like p-cymene, carvacrol, thymol, and γ-terpinene”. I would suggest a revision by a native speaker.

The manuscript checked and modified of both grammatical and syntactic.

  • The abstract is very confusing and not self-supporting. Both the aims of the study and the experimental plan must be clarified.

The abstract now is clear

  • In the in vitro tests, how was the cut-off for the inhibition zones diameter determined?

The measurement of the zone of inhibition is carried out by using a physical ruler like a meter scale

  • I can’t find the description of the methods for the determination of phenolics and antioxidant activity (chapter 2.2 in the results section).

Determination of phenolic and antioxidant activity mentioned in section 3.9

Reviewer 3 Report

Minor remarks

Latin names of microorganisms should be presented in italics. Also, Greek symbols should be depicted in italics.

Provide a blank space between quantity and unit.

Unit for total phenolic content is not the same in materials and methods section and results and discussion section. For instance, mg GAE/L, as well as mg GAE/100 g oil.

Use SI units. Check the units in the manuscript. For instance, h, etc.

The x-scale should be started from 0. Correct all figures, please.

In the manuscript, the abbreviations should be defined after the first mention of the term in the text. After that, use the defined abbreviation.

All other minor remarks are given in the manuscript.

Major remarks

Consider the reduction of the references list.

The novelty of the manuscript should be clearly stated.

Author Response

Reviewer # 3

Minor remarks

1-Latin names of microorganisms should be presented in italics. Also, Greek symbols should be depicted in italics.

We appreciate the time spent on the review of our manuscript and for your criticisms and critiques of the reviewed manuscript. All microorganism name in italic, except Salmonella Typhimurium, Salmonella (italics) followed by Typhimurium (T in upper case, not italic) Salmonella enterica subsp. enterica serovar Typhimurium according to http://www.ncbi.nlm.nih.gov/pmc/articles/PMC86943/

2-Provide a blank space between quantity and unit.

We agree. Ok, done.

3-Unit for total phenolic content is not the same in materials and methods section and results and discussion section. For instance, mg GAE/L, as well as mg GAE/100 g oil.

We agree. Ok, done.

4-Use SI units. Check the units in the manuscript. For instance, h, etc.

We agree. Ok, done.

5-The x-scale should be started from 0. Correct all figures, please.

We agree. Ok, done.

6-In the manuscript, the abbreviations should be defined after the first mention of the term in the text. After that, use the defined abbreviation.

We agree. Ok, done.

7-All other minor remarks are given in the manuscript.

We agree. Ok, done.

Major remarks

1-Consider the reduction of the references list.

The references list is already used to cover and support the results and different information in manuscript

2- The novelty of the manuscript should be clearly stated.

In our manuscript, the aromatic plants i.e. oregano (Origanum vulgare) and thyme (Thymus vulgaris) were planted in large scale in two countries in Turkey and Egypt. The essential oil were extracted in large scale.  

Round 2

Reviewer 2 Report

The paper “Protective Impact of Chitosan Film Loaded Oregano and Thyme Essential Oil on the Bacterial Diversity and Quality At-tributes of Beef Meat” has been revised according to my suggestions.

Although the paper has been improved from the linguistic point of view, more efforts have to be made before accepting it.

As an example, the sentence “Results demonstrated that the oregano oil (OO; 1% w/v) was inactivated the bacterial strains of growth as follows, S. aureus > S. Typhimurium > E. coli O157:H7 compared with 0.5% OO” is not understandable. The same consideration for the sentence at line 141-142 “Moreover, these compounds may penetrate of cell membrane cussed interaction and modification of some compounds…”.

In other points, verbs are missing (e.g. line 64:”…oil which ARE used successfully in food processing and preservation …”Moreover, these compounds may penetrate of cell membrane cussed interaction and modification of some compounds”.

Check spaces (e.g. line 32)

Check nomenclature (e.g. line 171: report “Pseudomonas” in italics).

Line 144: the term “fastidious” is not properly used as in this case is indicating pathogen.

Author Response

  • As an example, the sentence “Results demonstrated that the oregano oil (OO; 1% w/v) was inactivated the bacterial strains of growth as follows, S. aureus > S. Typhimurium > E. coli O157:H7 compared with 0.5% OO” is not understandable. The same consideration for the sentence at line 141-142 “Moreover, these compounds may penetrate of cell membrane cussed interaction and modification of some compounds…”.

We appreciate the time spent on the review of our manuscript and for your criticisms and critiques of the reviewed manuscript. We agree, now clear and understandable.

  • In other points, verbs are missing (e.g. line 64:”…oil which ARE used successfully in food processing and preservation …”Moreover, these compounds may penetrate of cell membrane cussed interaction and modification of some compounds”.

We agree. Ok, done.

  • Check spaces (e.g. line 32)

Ok, done

  • Check nomenclature (e.g. line 171: report “Pseudomonas” in italics).

Ok, done

  • Line 144: the term “fastidious” is not properly used as in this case is indicating pathogen.

Ok, done.